# Controversial Properties of Amyloidogenic Proteins and Peptides: New Data in the COVID Era

**DOI:** 10.3390/biomedicines11041215

**Published:** 2023-04-19

**Authors:** Andrei Surguchov, Fatemeh N. Emamzadeh, Mariya Titova, Alexei A. Surguchev

**Affiliations:** 1Department of Neurology, University of Kansas Medical Center, Kansas City, KS 66160, USA; 2Analytical Development Department, Iovance Biotherapeutics, Inc., Tampa, FL 33612, USA; 3The College of Liberal Arts & Sciences, Kansas University, Lawrence, KS 66045, USA; 4Department of Surgery, Section of Otolaryngology, Yale School of Medicine, Yale University, New Haven, CT 06520, USA

**Keywords:** amyloidogenic proteins, amyloidogenic peptides, α-synuclein, β-amyloid, Parkinson’s disease, Alzheimer’s disease, COVID-19, SARS-CoV-2, amyloidosis

## Abstract

For a long time, studies of amyloidogenic proteins and peptides (amyloidogenic PPs) have been focused basically on their harmful properties and association with diseases. A vast amount of research has investigated the structure of pathogenic amyloids forming fibrous deposits within or around cells and the mechanisms of their detrimental actions. Much less has been known about the physiologic functions and beneficial properties of amyloidogenic PPs. At the same time, amyloidogenic PPs have various useful properties. For example, they may render neurons resistant to viral infection and propagation and stimulate autophagy. We discuss here some of amyloidogenic PPs’ detrimental and beneficial properties using as examples beta-amyloid (β-amyloid), implicated in the pathogenesis of Alzheimer’s disease (AD), and α-synuclein—one of the hallmarks of Parkinson’s disease (PD). Recently amyloidogenic PPs’ antiviral and antimicrobial properties have attracted attention because of the COVID-19 pandemic and the growing threat of other viral and bacterial-induced diseases. Importantly, several COVID-19 viral proteins, e.g., spike, nucleocapsid, and envelope proteins, may become amyloidogenic after infection and combine their harmful action with the effect of endogenous APPs. A central area of current investigations is the study of the structural properties of amyloidogenic PPs, defining their beneficial and harmful properties, and identifying triggers that transform physiologically important amyloidogenic PPs into vicious substances. These directions are of paramount importance during the current SARS-CoV-2 global health crisis.

## 1. Introduction

Amyloidogenic PPs are involved in many human diseases, including AD, PD, and other neurodegenerative disorders characterized by the buildup of proteinaceous aggregates predominately in the brain. Several amyloidogenic PPs are involved in the pathogenesis of cardiac, renal, gastrointestinal amyloidosis, systemic amyloidosis, and other disorders. Thus, dozens of amyloidogenic PPs with various structures may be considered as culprits responsible for many human diseases. Amyloidosis is a broad name for diseases characterized by the accumulation of structured oligomers and amyloid fibrils in cells and tissues, causing organ dysfunction and, sometimes, death [1,2]. Amyloid oligomers are supposed to be more toxic than fibrils [1,2,3]. The number of recognized amyloidogenic PPs, as well as insoluble amyloid fibrils originating from soluble precursor proteins, has increased over time. It is common knowledge that amyloidogenic PPs are prone to aggregate, forming various structures with a fibrillar morphology associated with human diseases. Prone-to-aggregate amyloidogenic PPs may misfold, lose their normal organization and function, and form intracellular or extracellular deposits and inclusions with a high content of β-sheet-rich fibrillar structures [1,2]. The accumulation of such inclusions causes a vicious cycle of cellular damage that leads to inflammation, apoptosis, neurodegeneration, or cell death. Due to the fast aging of the population worldwide and the advancement of diagnostic methods, the number of identified illnesses associated with the buildup of amyloid proteins has risen [3,4].

Amyloids are usually long, unbranched fibrous protein molecules formed through the templated polymerization of thousands of monomeric peptides. The fibers are 5–15 nm wide and several micrometers long. They bind the dye Congo red and display fluorescence birefringence after binding to the fluorescent dye thioflavin T (ThT)—a cationic benzothiazole dye used to analyze nucleation-dependent polymerization. Thanks to the application of advanced technologies with higher resolution, it has become clear that, on the molecular level, amyloids comprise a wide diversity of structures [5].

Particularly surprising has been the finding that identical polypeptides can fold into multiple, distinct amyloid conformations. A subclass of amyloids in which protein aggregation is self-propagating and infectious is called prions; this structural diversity can lead to distinct heritable prion states or strains [5]. It is not completely clear why, given its harmful effects, evolution has preserved amyloidogenic PPs in the cell. There is a hypothesis that the evolutionary conservation of amyloidogenic PPs is due to their beneficial properties, which have not been deeply investigated and sometimes remain unknown. The growing number of results demonstrating the deleterious effects of amyloidogenic PPs overshadow the data about their beneficial properties. Here we try to balance the evidence showing the useful effects of amyloidogenic proteins with the bulk of data reporting their disease-causative neurotoxicity. It is important to detect the mechanism of conversion of beneficial amyloidogenic PPs into toxic substances before the disease symptoms become evident. Such detection would allow beginning treatment at the earliest step of the disease when it may be the most efficient.

## 2. Pathogenic Properties of Amyloidogenic PPs

### “Friends May Come and Go, but Enemies Accumulate” Thomas Jones

For many years amyloidogenic PPs (which can be called “amyloids”) were the focus of attention of biochemists, biophysicists, and clinicians because of their role in many human diseases called amyloidoses [2,3,4,5,6]. The accumulation of amyloidogenic PPs is associated with the deposition of misfolded proteins in tissues, causing organ damage. However, before accumulation, misfolding, and aggregation, monomeric amyloidogenic PPs are not toxic and may exert physiological functions. Several types of amyloidoses are known, including hereditary, sporadic, systemic, and organ-specific forms of the disease.

More than 40 various globular, soluble proteins may undergo misfolding and aggregation leading to the formation of insoluble fibrils [4]. Amyloid deposits are formed from globular, soluble proteins, which undergo misfolding and aggregation in response to overexpression, proteolytic digestion, mutations, etc., releasing amyloidogenic peptides. Some wild-type proteins have an intrinsic propensity to misfold and aggregate [6]. The polypeptide chains generally form β-sheet structures that aggregate into long fibers; however, identical polypeptides can fold into multiple distinct amyloid conformations [7,8]. Misfolded protein aggregates can self-propagate based on seeding and spread the pathology between tissues and cells in a way analogous to the action of infectious prions in prion diseases [8].

A majority of amyloidogenic proteins are intrinsically disordered. Amyloidogenic PPs can be totally unstructured or contain structured domains and long stretches of inherently disordered regions. These regions are primarily made of polar or charged amino acids, lacking sufficient quantity of hydrophobic residues that mediate cooperative folding. The conformations adopted are affected mainly by amino acid sequence, amino acid motifs, and charge distribution [9,10,11]. Amyloidogenic proteins undergo misfolding, altering their native states to form β-sheet-rich structures, ranging from small oligomers to large fibrillar aggregates associated with diseases [9,12].

Intrinsically disordered proteins (IDPs) are highly prevalent in many proteomes, including that of humans; they play an important role in cellular processes such as the regulation of transcription and translation [9,10], cell cycle control [11,12], and cell signaling [12,13]. Changes in the cellular milieu and/or mutation(s) in IDPs can disrupt normal protein functions, resulting in misfolding and aggregation/ fibrillation [14,15]. Additional information about the pathological properties of amyloidogenic PPs can be found in excellent reviews [4,5,6].

The pathogenic properties of host amyloidogenic PPs may be strengthened after interaction with viral proteins. For example, β-amyloid Aβ1-42 binds to various viral proteins, e.g., with the spike protein S1 subunit (S1) of SARS-CoV-2 (Figure 1), and the viral receptor, angiotensin-converting enzyme 2 (ACE2) [16]. Importantly, Aβ1-42 reinforces the binding of the S1 of SARS-CoV-2 to ACE2 and enhances the viral entry and production of IL-6 in a SARS-CoV-2 pseudovirus infection model. These findings emphasize the critical role of Aβ1-42 in increasing SARS-CoV-2 intrusion and suggest mechanisms by which Aβ1-42 enhances SARS-CoV-2 infection or inflammation [16].

Another member of amyloidogenic PPs, α-synuclein [17,18], also directly interacts with SARS-CoV-2 proteins, i.e., the spike (S) and nucleocapsid (N) proteins [19]. Recent data show that the expression of α-synuclein is upregulated, and its aggregation is enhanced by viral S- and N-proteins. Importantly, SARS-CoV-2 proteins cause Lewy-like pathology in cells overexpressing α-synuclein. More data about the amyloidogenic properties of ARS-COVID proteins are presented in Section 5, “Amyloidogenic ARS-COVID proteins”.

## 3. Physiological Roles of Amyloidogenic PPs

### “You Never Really Know Your Friends from Your Enemies until the Ice Breaks.” Eskimo Proverb

The implication of amyloids as pathogens in many illnesses has been confirmed by an overwhelming amount of proof that has overshadowed the data about their physiological role, which is vital for the living cell. The concept of amyloids as vicious pathogens responsible for dozens of fatal human diseases was so dominating that it slowed down studies of so-called “functional amyloids” and their roles in various activities. However, the accumulating data about the involvement of amyloids in normal physiological functions compelled us to combine the efforts of multi-disciplinary scientists and develop a branch of science called “amyloidomics” [20,21].

In many later studies, amyloids have been proven to serve various biological functions beyond the development of pathological processes [3]. Since the beginning of the 21st century, researchers have reported much evidence that amyloids perform specific functional roles [3,13,14]. In many cases, monomeric amyloidogenic PPs are not toxic and are involved in various important physiological functions, but once they aggregate into amyloid fibrils, they become toxic to cells and tissues.

The functional amyloids have been detected in both prokaryotes and eukaryotes [3,9,10]. For instance, in Escherichia coli, amyloids contribute to the formation of biofilms [15]. Amyloids have been found to be functional in many aspects, as structural components in bacteria and viruses, as biochemical regulators functioning as hemostatic agents in human beings, and as scaffolds, or molecular chaperones; they also play a role in sexual reproduction [3,10,19,20,21,22]. Amyloids are involved in melanin formation in the melanosomes of human skin cells [22].

A detailed, complicated mechanism of the transformation of physiologic proteins into toxic amyloid is described for the premelanosome transmembrane glycoprotein (PMEL) protein [23,24]. This mechanism also includes proteolytic digestion, but the initial step of such a transition is the three amino-acid insertion in the PMEL transmembrane domain. The insertion, called the Dominant White (DW) mutation, causes the formation of the abnormal compact fibrillar structures of the PMEL protein. It leads to abnormal oligomerization of the transmembrane domain resulting in the formation of increased numbers of toxic oligomers or the creation of anomalous fibril polymorphs that are toxic to the cells. In the final stage, this transition is mediated by several proteases [23,24].

The physiological role of various amyloidogenic PPs was discovered in all taxonomic groups [13,14,25,26]. They possess trophic, neuroprotective properties [27,28], antioxidant [29] and antimicrobial activity [30], and other physiologically important properties. For example, transmembrane amyloid β-precursor protein (APP), the precursor of the β-amyloid peptide, is required for normal synaptic function [31], dendritic spines remodeling, synaptic homeostasis, and molecular pathways of neurotransmission [32,33,34,35,36].

An increasing amount of data has indicated that APP and its cleavage products play vital roles in regulating intracellular processes [37]. Full-length APP directly regulates metabolism in the CNS and peripheral tissue and can modulate mitochondrial functions [38,39,40]. It can be cleaved by proteases in different ways to produce a variety of short peptides, some of which, e.g., β-amyloid, possess toxic properties [40,41].

Aβ peptides are produced from the precursor protein after digestion by the beta-cite APP cleaving enzyme (BACE1) and subsequent digestion by γ-secretase. They regulate synaptic function and, in addition, possess protective properties against injury and infection, and participate in repairing leaks in the blood-brain barrier [42]. Importantly, Aβ peptide synthesis rapidly rises in response to physiological stress and usually reduces upon recovery [42]. Under physiological conditions, Aβ production is reportedly associated with neuronal activity [43] and regulated by the sleep/wake cycle [44]. A deficiency in endogenous Aβ causes synaptic dysfunction and cognitive defects, whereas a mild increase enhances long-term potentiation and leads to neuronal hyperexcitability [41].

An essential physiological role of amyloids is their participation in regulating RNA processing and degradation and controlling gene expression on the transcriptional and translational levels [45]. For example, the functional amyloid hnRNPDL protein forms stable amyloid fibrils and, at the same time, can bind RNA by RNA-binding domains located as a solenoidal amyloid coat around the core. This nuclear RNA-binding protein hnRNPDL is involved in transcription and RNA processing and forms ordered amyloid fibrils under physiologic-like conditions [45].

Adding to the experimental data points for the protective function of Aβ, it may play a role in the protective response to age-related metabolic pressures in the cell [42], cytoprotective pathways, and intracellular signaling [32].

The conversion of physiologically important amyloidogenic PPs to toxic proteins and peptides depends on the influence of various circumstances. One of the factors causing the formation of misfolded proteins and amyloidogenic inclusions leading to pathology is the overexpression and accumulation of amyloidogenic PPs. Usually, such conversion begins when previously healthy proteins start to accumulate over a certain level, lose their normal structure and physiological functions (misfolding, proteolytic digestion, post-translational modifications, etc.), and form fibrous deposits within and around cells. Such protein misfolding and deposition processes may disrupt the healthy function of tissues and organs. One of the triggers initiating this conversion may be the overexpression of APP and its accumulation over a certain limit. Another reason may be the proteolytic digestion of a precursor protein, as happens with APP, causing its conversion into Aβ [4,5,6,7,21,22,25].

It is vital to understand how and why amyloidogenic PPs possessing some important physiological activity and beneficial properties are converted to harmful substances associated with human diseases. Such understanding will allow us to find treatments for these fatal diseases. In addition to the physiological properties mentioned here, we will discuss the antiviral and antimicrobial properties of amyloidogenic PPs in a separate Section 4 below.

## 4. Antimicrobial and Antiviral Properties of Amyloidogenic PPs

Numerous results point to the ability of α-synuclein, β-amyloid, and other amyloidogenic PPs to acquire toxic properties [1,2,3,4,17,44]. These findings have been so convincing that data about their antibacterial and antiviral activity has remained in the shadow for a long time. Meanwhile, such data were accumulating and becoming more and more compelling. The antibacterial, antimicrobial, and antiviral activity of amyloidogenic PPs are described in many comprehensive publications [45,46,47,48,49,50], so here we will briefly mention only several examples illustrating the antimicrobial and antiviral properties of Aβ and α-synuclein.

### 4.1. Antimicrobial Activity of Aβ Peptide and α-Synuclein

Aβ peptide. Various experimental results support the antimicrobial properties of Aβ peptide and α-synuclein in vivo. For example, Aβ exerts antimicrobial activity against eight common and clinically relevant microorganisms [28]. The authors compared the antimicrobial activity of Aβ and cathelicidin LL-37, a multitasking antimicrobial peptide, and found that their potency is equivalent. In some cases, Aβ had an even greater antimicrobial potency than LL-37. These results demonstrating powerful Aβ antimicrobial activity suggest that they may normally function in the innate immune system [28]. Due to their protective properties, Aβ and its derivatives are considered important and potent natural antimicrobial peptides. They are even called “clinical antibiotics”, which can potentially be developed into clinically useful agents [50,51].

Kumar et al. (2016) [51,52] demonstrated that amyloid-β protects against microbial infection in 5XFAD mouse and transgenic nematode C. elegans models of Alzheimer’s disease. They found that a higher amyloid-β expression correlated with better host survival whereas low expression was associated with higher mice mortality.

In another study examining amyloid-β protective activity Eimer and coauthors (2018) [53], using an AD mouse model and neuronal cell cultures, found that Aβ peptide protects against Herpesviridae. The results suggest that Aβ peptide might play a protective role in CNS innate immunity.

Readhead et al. (2018) [54] constructed multiscale networks of the late-onset Alzheimer’s disease (AD)-associated virome from human post-mortem tissue. This study revealed pathogenic regulation of neuropathological, molecular, and clinical networks by common viruses, including herpesvirus 6A and 7. Another important observation was finding regulatory connections linking viral abundance and modulators of APP processing, e.g., the induction of APBB2, BACE1, and PSEN1 by HHV-6A [54].

Investigation of the herpes simplex virus type 1 (HSV1) role in Alzheimer’s disease pathogenesis revealed the striking colocalization of viral DNA and amyloid plaques, suggesting that it might be entombing the virus, thereby preventing viral replication [55].

α-Synuclein. Park et al. [56] showed that α-synuclein possessed activity against several species of bacteria, including Escherichia coli and Staphylococcus aureus. The authors isolated the human α-synuclein gene from the cDNA library, expressed the recombinant protein in E.coli, purified it on agarose affinity gel, and assayed its antibacterial and antifungal activity. Although the characteristics of α-synuclein used in these assays were not described, most probably, the recombinant protein was not aggregated. The authors also proved α-synuclein antifungal activity against Aspergillus flavus, Aspergillus fumigatus, and Rhizoctonia solani. The authors assumed that Aβ and α-synuclein antimicrobial activity might be explained by the similarity of their general properties, e.g., membrane binding affinity, the ability to induce an innate immune response, etc. [56].

Convincing data pointing to the antibacterial effects of α-synuclein have been gathered using α-synuclein knockout mice (α-syn KO) [57]. α-Syn KO mice had substantially fewer immune cells migrating into the peritoneal cavity after the immune challenge. Furthermore, it was impossible to induce peritonitis in the α-syn KO mice [57].

Interestingly, adult mice with knockout of all three synuclein genes (α, β, and γ) display no overt phenotype, normal development, and usual neurological and behavioral characteristics [58,59]; however, they exhibit deficient host defense against infectious agents [47,57]. These findings confirm synuclein’s role in protecting against infectious diseases and their role in the mammalian immune system.

In another study, α-syn KO mice succumbed to viral encephalitis more easily than their wild-type littermates used as control [60]. In a bacterial sepsis model, α-syn KO mice were less capable of controlling infection after the intravenous injection of Salmonella typhimurium [60]. These findings add new evidence about the role of endogenous α-synuclein in innate immune defense. The results of another study show higher mortality in the α-syn KO mice to two types of neurotropic RNA viruses: the West Nile virus (WNV) and the Venezuelan equine encephalitis virus (VEEV) [61]. The results of this study demonstrate that native α-synuclein expression in neurons inhibits viral growth.

### 4.2. Antiviral Properties of Aβ and α-Synuclein

Aβ peptides. In addition to antibacterial properties, Aβ peptides also possess antiviral activity against several types of viruses. The molecular mechanism of Aβ antiviral activity toward herpes simplex virus 1 (HSV-1) is mediated by the interaction of Aβ with HSV-1 protein gB. This interaction causes the impairment of HSV-1 infectivity by preventing the virus from fusing with the plasma membrane [62,63].

### 4.3. Antiviral Activity of α-Synuclein

The interaction of α-synuclein with SARS-CoV-2 is rather complex [64]. On the one hand, α-synuclein overexpression restricts SARS-CoV-2 neuroinvasion and reduces the degeneration of dopaminergic neurons. On the other hand, the SARS-CoV-2 virus speeds up α-synuclein aggregation [64].

The antimicrobial and antiviral activity of α-synuclein may be explained by its immune regulatory function and ability to regulate the release of pro-inflammatory cytokines [65] and facilitate immune responses against different infections [57,66].

α-Synuclein is able to restrict viral replication and prevent virus-induced neuronal damage in experimental mice [61]. This effect presumably occurs due to the alteration of the membrane transport from the ER to the Golgi body [67].

There are several pathways involved in the mechanism of α-synuclein antiviral activity. When α-synuclein localizes to ER-derived membranes, it modulates virus-induced ER stress signaling and inhibits viral replication, growth, and injury in the CNS. The reduced expression of α-synuclein decreases the phagocytic activity of microglia and response to the pathogenic agents [68]. These properties ensure α-synuclein’s neuroprotective effect. Overexpression of α-synuclein in PD patients may limit SARS-CoV-2 neuroinvasion and the degeneration of dopaminergic neurons [69]. Recently, it has been reported that α-synuclein can stabilize the RG-1 gene of SARS-CoV-2 involved in the synthesis of nucleocapsid protein (N-protein), which affects the cellular process during replication [70].

Monogue et al. (2022) [71] further investigated molecular mechanisms underlying antiviral α-synuclein activity, including its effect on induced immune responses to viral infections in the brain. The researchers challenged α-syn KO mice and human α-syn KO dopaminergic neurons with RNA virus infections. They found that α-synuclein was needed for the expression of interferon-stimulated genes in neurons. Importantly, after the treatment with type 1 interferon human α-syn KO neurons lost the ability to induce a wide range of interferon-stimulated genes. These results suggest that α-synuclein cooperates with type 1 interferon signaling. Furthermore, after interferon treatment α-synuclein accumulates in the nucleus of human neurons and modulates interferon-mediated phosphorylation of STAT2. Activated STAT2 co-localizes with α-synuclein after type 1 interferon stimulation. These findings demonstrate that α-synuclein expression endorses interferon responses by localizing to the nucleus, mediating STAT2 activation, co-localizing with phosphorylated STAT2 in neurons, and maintaining expression of interferon-stimulated genes. These data provide a novel mechanism that links interferon activation and α-synuclein functions in neurons [71].

## 5. Amyloidogenic ARS-COVID Proteins

The accumulating data demonstrates that some of the symptoms of amyloidogenic PPs-related diseases resemble COVID-19 signs [72]. The SARS-CoV-2 infection has been associated with severe neurological symptoms. Furthermore, the analysis of viral proteins in infected cells has revealed that some of them possess amyloidogenic properties, e.g., nucleocapsid (N) protein, spike (S) protein (S-protein or homotrimeric surface spike protein), and envelope (E) protein [72] (Figure 1). Importantly, these viral proteins, in addition to being amyloidogenic themselves, may enhance the amyloidogenesis of host proteins. Thus, after viral infection an interaction between endogenous and viral proteins in the host cells synergistically increases their amyloidogenic potential.

The amyloidogenic properties of COVID-19 proteins, i.e., S, N, and E, have been confirmed by biochemical and biophysical examination [72]. Noteworthy is that two of them, the N- and S-proteins, are the most abundant in COVID-19-infected human cells.

These viral proteins together with host proteins are responsible for various syndromes following COVID-19 infection, including Parkinsonism in young COVID patients, acute respiratory distress syndrome (ARDS), and cardiovascular disorders issues. Amyloidogenic SARS-CoV-2 proteins can also be responsible for some of the neurodegenerative complications following infection [73]. Some resemblance between COVID-19 and amyloidosis symptoms point to possible similarities in their pathogenic mechanisms [74,75].

Several outcomes of COVID-19 infection are remarkably similar to the symptoms of systemic AA amyloidosis (AA here stands for amyloid A). This type of amyloidosis is characterized by the abnormal aggregation of the serum amyloid A (SAA) protein [76]. There is an interesting hypothesis according to which AA amyloidosis is a factor causing systemic pathologies after a severe form of coronavirus disease [77]. The course of the pathology is aggravated, and it spreads quickly because SAA overproduction triggers the formation of amyloid aggregates that are deposited on the walls of blood vessels in various organs and tissues, causing inflammation of the affected tissues and vascular thrombosis [76].

Several amyloidogenic proteins are found in the proteome of SARS-CoV, including fragments of two structural proteins, i.e., the C-terminal end and transmembrane domain of the envelope protein (E-protein) and the membrane protein (M-protein). Amyloidogenic proteins NSP4 and NSP6 were also found among non-structural viral proteins [24].

Analysis of the open reading frames of the SARS-CoV-2 proteome for amyloidogenic sequences using computational methods identified two prone to aggregate amino acid sequences (ORF6 and ORF10) in addition to known viral proteins [78].

Spike peptide S-191 from the SARS-CoV-2 S-protein with amino acid sequence FVFKNIDGYFKIYSKHTPIN also possesses high amyloidogenic properties. The S-191 peptide is released from the S-protein by endoproteolysis catalyzed by neutrophil esterase and other immune-responsive proteases. This digestion occurs soon after the infection [79,80]. Importantly, N-protein enhances protein aggregation causing the production of stable fibrillar morphology. This upsets the endogenous α-synuclein proteostasis, impedes normal protein balance, and may cause Parkinsonism [72].

An important supplement to our understanding of how SARS-CoV-2 viral proteins might contribute to the pathogenesis of Parkinson’s disease is the results of a study demonstrating that the interaction of the SARS-CoV-2 nucleocapsid N-protein and α-synuclein accelerates amyloid formation [80]. Moreover, the N-protein stimulates the generation of multiprotein complexes and ultimately promotes the formation of amyloid fibrils. Experiments with the microinjection of the N-protein in SH-SY5Y cells showed that the N-protein damaged the α-synuclein proteostasis and enhanced cell death. These results suggest the existence of direct interactions and cross-seeding (Figure 2) between the N-protein of SARS-CoV-2 and α-synuclein which serve as the molecular basis for the association between SARS-CoV-2 infections and Parkinson’s disease [80]. Currently, various models have been proposed [81] and further studies are needed to clarify the exact mechanism of cross-seeding.

SARS-CoV-2 proteins interact with the negatively charged C-terminal region of α-synuclein which speeds up the misfolding and fibrosis of α-synuclein [80,81]. According to theoretical bioinformatic prediction, the S-protein contains several amyloidogenic amino acid sequences. Isolated peptides corresponding to these sequences are able to aggregate and form amyloid fibrils, confirmed by three independent criteria: (1) Nucleation-dependent polymerization kinetics by thioflavin T (ThT)—a cationic benzothiazole dye used to analyze nucleation-dependent polymerization. (2) Amyloid detection with Congo red staining. (3) Ultrastructural fibrillar morphology characteristics. The 20-amino-acid-long amyloidogenic peptides are located at positions 192–211, 601–620, and 1166–1185 of the S-protein [82].

The S-protein produces amyloid-like fibrils after in vitro incubation with the protease neutrophil elastase, including very amyloidogenic spike peptide located between amino acids 194 and 203. After infection by SARS-CoV-2, similar peptides may be produced by the endoproteolytic digestion of S-protein in vivo. Protease neutrophil elastase efficiently cleaves S-protein, rendering exposure of the amyloidogenic segments and accumulation of the amyloidogenic peptide 194–203, part of the most amyloidogenic synthetic spike peptide [82].

## 6. Relationship between COVID-19 Viral Proteins and Amyloidogenic PPs

A significant number of patients infected by SARS-CoV-2 have neurological symptoms involving dizziness, difficulty concentrating, loss of taste and smell, seizures, reduced alertness, etc. SARS-CoV-2 infection also affects the brain’s dopaminergic system, presumably through systemic inflammatory responses, and is associated with alterations in the gut microbiome. Thus, some of the neurological symptoms found in COVID-19 patients are similarly frequent in patients with Parkinson’s disease and other neurodegenerative disorders. Furthermore, various biochemical pathways, such as oxidative stress, protein aggregation, and inflammation, display resemblances between COVID-19 and Parkinson’s disease. Therefore, a detailed comparison of symptoms in patients with Parkinson’s disease and individuals after SARS-CoV-2 reveals many parallels and intersections between these two disorders [83]. The association between SARS-CoV-2 and synucleinopathies might be realized via the effect of viruses on α-synuclein, the expression of which is upregulated after viral infection.

Both the S- and N-proteins of SARS-CoV-2 increase α-synuclein expression, causing Lewy-like pathology [19]. Moreover, the S1 protein of SARS-CoV-2 binds several aggregation-prone proteins, including Aβ and α-synuclein (Idrees). Amyloid formation of α-synuclein is accelerated by the SARS-CoV-2 N-protein. Wu et al. found that α-synuclein had a higher binding affinity to the SARS-CoV-2 S-protein and N-protein [19].

The S1 protein of SARS-CoV-2 can bind several aggregation-prone proteins, including Aβ and α-synuclein. α-Synuclein has higher binding affinity to SARS-CoV-2 S1 protein [84]. Amyloid formation of α-synuclein is accelerated by the SARS-CoV-2 N-protein, suggesting a role of this viral protein in the pathogenesis of synucleinopathies.

Two probable mechanisms explaining how the virus can catalyze aggregation of the aggregation-prone brain proteins have been described [85]. According to the first mechanism, seeding protein aggregation on intact viral particles is carried out by spike proteins. The second mechanism proposes that a peptide derived from the spike protein acts as a functional amyloid to cross-seed aggregation of brain proteins. In both of these scenarios SARS-CoV-2 serves as a seed to accelerate the aggregation of brain proteins. The authors conclude that targeting the interaction of viral particles with the brain proteins might be an appropriate way to lower these cross-seeding and aggregation events.

An interesting hypothesis is that SARS-CoV-2 proteins are carried out as exosomal cargo together with host proteins, promoting neurodegenerative and neuroinflammatory cascades, leading to the development of synucleinopathies [86].

## 7. Conclusions

Protein folding provides the basis for life on our planet, ensuring the fulfillment of the major biological functions encoded by unique amino acid sequences. Protein misfolding is a source of many diseases, sometimes called conformational diseases [12]. Human protein aggregation and its association with diseases have been under investigation for decades and is relatively well studied. The accumulated data have revealed the fundamental aspects of amyloid fibril formation and its links to human disease [2,3,4,5,6]. These findings help to develop therapeutic strategies to combat this group of diseases. However, the mechanisms causing previously healthy proteins to lose their normal structure and form fibrous deposits within and around cells require further investigation.

It still needs to be completely understood why individual mechanisms exist for the transition of different proteins and peptides. Currently, the data about viral protein aggregation is more limited. The recent COVID-19 pandemic, caused by the SARS-CoV-2 virus, attracted attention to the mechanisms underlying viral infection including the study of viral protein amyloidoses and their interaction with host proteins.

New findings point to a delicate communication between the viral and host proteomes after infection, including the cross-seeding of host proteins by viral proteins [19,78,79,80,81,82,83,87,88].

Recent data demonstrating a direct communication between human α-synuclein and the N-protein of SARS-CoV-2 unveiled the molecular basis for the relations between virus infections and Parkinsonism [75]. Other effects of viral proteins on human cells have been recently discovered, for example, findings that the aggregation of viral proteins may assist the virus in seizing the replication machinery of a host cell and using it for its own purpose, further damaging the host cells. Moreover, viral infection initiates the misfolding and aggregation of host proteins by cross-seeding, increasing the damage to the host cells. These results agree with the existence of connections between neurodegenerative diseases and viral infection [89,90], although some mysterious associations still remain unsolved.

Further research will reveal these mysteries, because as Dan Brown wrote in his *The Da Vinci Code*: “Life is filled with secrets. You can’t learn them all at once.”

## Figures and Tables

**Figure 1 biomedicines-11-01215-f001:**
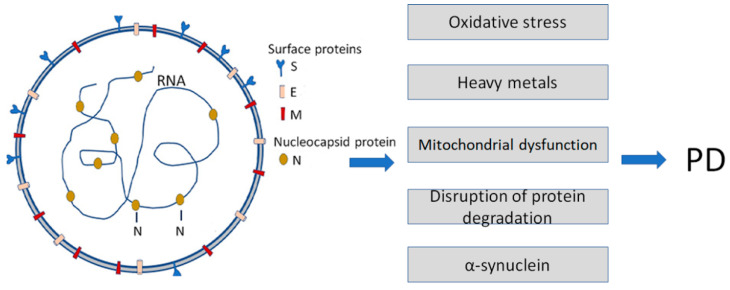
Infection with Severe Acute Respiratory Syndrome Coronavirus 2 (SARS-CoV-2) is a causative agent of COVID-19. It enhances some of the Parkinson’s disease (PD)-specific factors and pathogenic pathways contributing to PD development. SARS-CoV-2 particles (left) contain positive-sense single-stranded RNA with a bound nucleocapsid protein and surface proteins: S (spike), M (membrane), and E (envelope), inserted in the lipid bilayer.

**Figure 2 biomedicines-11-01215-f002:**
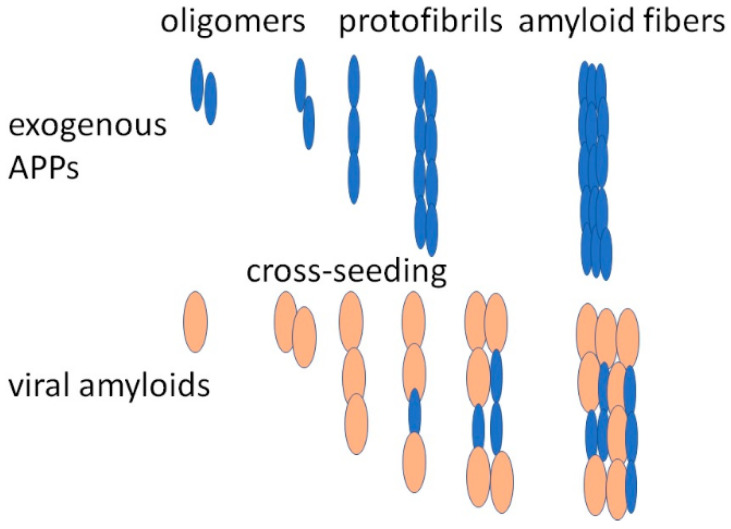
A schematic diagram of the cross-seeding of amyloidogenic PPs and viral amyloids. The presence of heterogeneous seeds accelerates amyloid.

## Data Availability

The data presented in this study are openly available in PubMed.

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
