# Peer review of "Controversial Properties of Amyloidogenic Proteins and Peptides: New Data in the COVID Era"

_biomedicines, 2023, doi:10.3390/biomedicines11041215_

Round 1

Reviewer 1 Report

Review article "Controversial properties of amyloidogenic proteins and peptides: new data in the COVID era" (Andrei Surguchov, Fatemeh N Emamzadeh, Alexei A. Surguchev) contains important, interesting, and relevant information, and can be recommended for publication. However, it should be noted that it is rather sloppily written. For this reason it should be rewritten and clarified, paying attention to the attention to accuracy and coherence. The main remarks are listed below.

1. In the text of the article, the authors mainly write about the toxicity of amyloidogenic protein fibrils and only a few times mention the toxicity of their oligomeric forms. It seems to me that in the very beginning of the article it is necessary to emphasize the paramount importance of oligomeric forms in the pathological action of amyloids in accordance with the modern concepts of the origin and development of amyloid neurodegenerative diseases.

2. The almost complete identity of the first phrases of sections 2 and 3 indicates the above-mentioned negligence in the preparation of the article. One of the phrases should be should be rewritten.

3. The phrase "Numerous results pointing to the toxic properties of α-synuclein, β- amyloid, and other APPs [1-4, 17, 42]" is also inaccurate. After all, it is not these proteins exhibit toxicity but in their specific forms.

4. It should also be clarified which forms of Aβ and α-synuclein have antimicrobial and antifungal properties in this section "Park et al. [48] showed that α-synuclein possessed activity against several species of bacteria, including Escherichia coli and Staphylococcus aureus. Furthermore, the authors proved α-synuclein antifungal activity against Aspergillus flavus, Aspergillus fumigatus and Rhizoctonia solani. The authors assumed that Aβ and α-synuclein antimicrobial activity might be explained by the similarity of their general properties, e.g., membrane binding affinity, the ability to induce an innate immune

response, etc. [48]". 

5. Placing section 5 between sections 4 and 6 violates the logic of the presentation. It should be moved to the beginning of the review. It could also be deleted. Why introduce subsections 3.1 and 5.1 if there are no other subsections.

6. Superficially described sections that deal with the most important aspects of the article. Citing own review articles [17, 18] is clearly not enough. There are about 60 articles on the relationship between α-synuclein and COVID-19 in the  Pubmed database (α-synuclein + COVID-19). Even the interaction of α-synuclein with S-protein there are already 5 articles (α-synuclein + COVID-19 + S-protein). And this interaction may be important for understanding the possible effects of COVID-19 vaccination. The authors should expand on this section, by analyzing the latest data in this area.

7. What does the acronym ARS-COVID mean? The authors introduce it without deciphering it and use it further, it is not commonly used and is redundant.

8. On page 7 in the sentence "Amyloidogenic SARS-CoV-2 proteins can also be responsible for some of the neurodegenerative complications following infection [68-70]" references 68 and 69 are completely irrelevant. 

9. On page 8, immediately after figure 2, the sentence "fiber development raises the probability of polymorphism in the fibril conformation [77]." hangs in the air and is unclear where it has left from.

10. On page 8, in the sentence "SARS-CoV-2 infection is accompanied by many injuries resembling amyloid-caused diseases, including blood coagulation and fibrinolytic disorders, neurological and cardiac complications suggesting the involvement of viral S-protein [78]" the cited article by Nystrom et al suggests a mechanism rather than describing complications of SARS-CoV-2 infection at all.

Author Response

We would like to thank the reviewer for thoroughly reviewing our manuscript and for critiques and suggestions. Below are our responses to criticism.   

Reviewer 1.

  1. In the text of the article, the authors mainly write about the toxicity of amyloidogenic protein fibrils and only a few times mention the toxicity of their oligomeric forms. It seems to me that in the very beginning of the article it is necessary to emphasize the paramount importance of oligomeric forms in the pathological action of amyloids in accordance with the modern concepts of the origin and development of amyloid neurodegenerative diseases.

Response: We already had the following sentence in Section 2, on page 2 “Many data point to the smaller, soluble misfolded oligomers as the true cause of neuro-degeneration [9].” 

In order to strengthen this statement we corrected and reinforced the following sentences in the Introduction: ”Amyloidosis is a broad name for diseases characterized by the accumulation of structured oligomers and amyloid fibrils in cells and tissues, causing organ dysfunction and, sometimes, death [1, 2]. Amyloid oligomers are supposed to be more toxic than fibrils [1-3].

2. The almost complete identity of the first phrases of sections 2 and 3 indicates the above-mentioned negligence in the preparation of the article. One of the phrases should be should be rewritten.

Response: Thank you for this comment. We deleted redundant text from section 3.

3. The phrase "Numerous results pointing to the toxic properties of α-synuclein, β- amyloid, and other APPs [1-4, 17, 42]" is also inaccurate. After all, it is not these proteins that exhibits toxicity but in their specific forms.

Response: Thank you for the suggestion. We modified the text as follows: "Numerous results point to the ability of α-synuclein, β- amyloid, and other APPs to acquire toxic properties [1-4, 17, 42]".

4. It should also be clarified which forms of Aβ and α-synuclein have antimicrobial and antifungal properties in this section "Park et al. [48] showed that α-synuclein possessed activity against several species of bacteria, including Escherichia coli and Staphylococcus aureus. Furthermore, the authors proved α-synuclein antifungal activity against Aspergillus flavus, Aspergillus fumigatus and Rhizoctonia solani. The authors assumed that Aβ and α-synuclein antimicrobial activity might be explained by the similarity of their general properties, e.g., membrane binding affinity, the ability to induce an innate immune response, etc. [48]". 

Response: We added the following text: ”The authors isolated human α-synuclein gene from cDNA library, expressed the recombinant protein in E.coli, purified it on agarose affinity gel and assayed its antibacterial and antifungal activity. Although the characteristics of α-synuclein used in these assays are not described, most probably, recombinant protein was not aggregated. 

5. Placing section 5 between sections 4 and 6 violates the logic of the presentation. It should be moved to the beginning of the review. It could also be deleted. Why introduce subsections 3.1 and 5.1 if there are no other subsections.

Response. As recommended by the reviewer, we deleted almost the whole section 5 except two fragments. The first begins with “The conversion of physiologically important…” and ends with “..precursor protein causing its conversion into Aβ [4-7, 21-23]”. This fragment we added to the end of Section 3. The second fragment from the previous Section 5 dealing with PMEL we moved to section 3, as recommended by Reviewer 2 in comment 2. We also deleted subsections 3.1 and 5.1.

6. Superficially described sections that deal with the most important aspects of the article. Citing own review articles [17, 18] is clearly not enough. There are about 60 articles on the relationship between α-synuclein and COVID-19 in the Pubmed database (α-synuclein + COVID-19). Even the interaction of α-synuclein with S-protein there are already 5 articles (α-synuclein + COVID-19 + S-protein). And this interaction may be important for understanding the possible effects of COVID-19 vaccination. The authors should expand on this section, by analyzing the latest data in this area.

Response. We followed this recommendation and expanded the text dealing with the interaction between COVID-19 proteins and host amyloids. Since the reviewer considers it as the most important aspect of the article, we presented it as a separate section 6 “Relationship between COVID-19 viral proteins and amyloidogenic PPs.” We included in this section the most important articles mentioned in the Reviewer’s 1 comments. As a further step toward the expansion of this section, we added some additional information to Figure 1.

7. What does the acronym ARS-COVID mean? The authors introduce it without deciphering it and use it further, it is not commonly used and is redundant.

Response. We corrected it to SARS-COVID.

8. On page 7 in the sentence "Amyloidogenic SARS-CoV-2 proteins can also be responsible for some of the neurodegenerative complications following infection [68-70]" references 68 and 69 are completely irrelevant. 

Response. We replaced references 68 and 69 with [73].

9. On page 8, immediately after figure 2, the sentence "fiber development raises the probability of polymorphism in the fibril conformation [77]." hangs in the air and is unclear where it has left from.

Response. We replaced this sentence with the following: ”SARS-CoV-2 proteins interact with negatively charged C-terminal region of α-synuclein which speeds up the misfolding and fibrosis of α-synuclein [77].

10. On page 8, in the sentence "SARS-CoV-2 infection is accompanied by many injuries resembling amyloid-caused diseases, including blood coagulation and fibrinolytic disorders, neurological and cardiac complications suggesting the involvement of viral S-protein [78]" the cited article by Nystrom et al suggests a mechanism rather than describing complications of SARS-CoV-2 infection at all.

Response. We deleted this sentence. The mechanism proposed by Nystrom et al. remains at the end of the same paragraph. 

Reviewer 2 Report

In this review manuscript by Surguchov et al., the authors summarize possible of antiviral and antimicrobial roles of amyloidogenic proteins associated with neurodegenerative diseases such as amyloid β (Aβ) in Alzheimer’s disease and α-synuclein in Parkinson’s disease. These peptides/proteins are generally thought to be detrimental to synapses and neural cells which eventually results in neuronal loss and dementia. Considering the recent pandemic of COVID-19, the topic of this manuscript is timely. The manuscript is well-organized, however, needs to be revised.

1. Please do not use the term “APP” for “amyloidogenic proteins and peptides”. This is very confusing, because in the filed of neurodegenerative diseases and neuroscience, the term “APP” stands for “amyloid β precursor protein” that is sequentially cleaved by β- and γ-secretases to produce Aβ peptides. Also, the gene APP encodes Amyloid Beta Precursor Protein. Please see https://www.ncbi.nlm.nih.gov/gene/351.

2.The section 1, please introduce PMEL in melanosomes as an example of functional amyloids, although the authors referred to PMEL later.

3. On page 3, “APPs which can be called amyloids”…, there may be a confusion in the use of the term “amyloidogenic proteins and amyloid”, because in many cases, monomeric amyloidogenic proteins/peptides are not toxic and may exert physiological functions, but once they aggregate into amyloid fibrils, they become toxic to cells and tissues. Please specify the aggregation statues in each section throughout the manuscript.

4. On page 4, BACE stands for “beta-cite APP cleaving enzyme”, not “cleaving enzyme”.

5. On page 4, “Importantly, Aβ peptide synthesis rapidly rises in response to physiological stress and usually reduces upon recovery.” needs a ref. Under physiological conditions, Aβ production reportedly associated with neuronal activity (Ciritto et al, NEURON, 2005 etc.) and regulated by the sleep/wake cycle (Kang et al., SCIENCE, 2009 etc.).

6. 4.1. Antimicrobial activity of Aβ peptide and α-synuclein, please refer to the studies regrading Herpes virus, Aβ, and AD, such as Kumar, et al., SciTranslMed, 2016; Eimer et al., NEURON, 2018; Readhead et al., NEURON, 2018 etc., which is also reviewed in Itzhaki et al., FasebJ, 2017.

7. On page 5, regarding α-syn and antimicrobial activity, please specify the aggregation status of α-syn, because many amyloid fibrils, not monomers, are capable of disturbing membranes. “The results of this study demonstrate that native α-synuclein expression in neurons inhibits viral growth and injury in the CNS.”, “injury in the CNS” is not sure from the references.

8. On page 6, “either producing soluble amyloid precursor protein-α by α-secretase or generating shorter Aβ species such as Aβ1-15 and Aβ1-16 by the subsequent cleavage with β-secretase and α-secretase.”, the α-cut of APP and the subsequent γ-cut produces p3 peptides.

9. Figure 2, the scheme of “cross-seeding” may be misleading. Currently, various cross-seeding models have been proposed and it is not sure viral amyloids cross-seed oligomers of amyloidogenic proteins/peptides. Please cite appropriate references for the cross-seeding model. In the figure, the term “exogenous” may be confusing.

Author Response

  1. Please do not use the term “APP” for “amyloidogenic proteins and peptides”. This is very confusing, because in the filed of neurodegenerative diseases and neuroscience, the term “APP” stands for “amyloid β precursor protein” that is sequentially cleaved by β- and γ-secretases to produce Aβ peptides. Also, the gene APP encodes Amyloid Beta Precursor Protein. Please see https://www.ncbi.nlm.nih.gov/gene/351.

Response: To avoid confusion, we changed this abbreviation. Instead of “amyloidogenic proteins and peptides (APPs) now we use the following abbreviation: amyloidogenic PPs throughout the whole manuscript.

Due to these replacements, we now are able to use common abbreviation for “Amyloid Precursor Protein” which we abbreviate as APP.

  1. The section 1, please introduce PMEL in melanosomes as an example of functional amyloids, although the authors referred to PMEL later. We

Response: As recommended, we moved texts describing PMEL to Section 3, after reference [22] as an example of functional amyloids.

  1. On page 3, “APPs which can be called amyloids”…, there may be a confusion in the use of the term “amyloidogenic proteins and amyloid”, because in many cases, monomeric amyloidogenic proteins/peptides are not toxic and may exert physiological functions, but once they aggregate into amyloid fibrils, they become toxic to cells and tissues. Please specify the aggregation statues in each section throughout the manuscript.

Response: In response we added to the Section 3, page 4 the following sentence “In many cases, monomeric amyloidogenic PPs are not toxic and exert important physiological functions, but once they aggregate into amyloid fibrils, they become toxic to cells and tissues.” We also added to the beginning of section 2 the following text :”However, before accumulation, misfolding and aggregation monomeric amyloidogenic PPs are not toxic and may exert physiological functions. Similar explanations are used in other places of the manuscript. 

  1. On page 4, BACE stands for “beta-cite APP cleaving enzyme”, not “cleaving enzyme”. Response. Thank you, we made this change on page 4.
  2. On page 4, “Importantly, Aβ peptide synthesis rapidly rises in response to physiological stress and usually reduces upon recovery.” needs a ref.

Response. Thank you, we added reference on Brothers et al. 2018 [40], Ciritto et al., 2005 and Kand 2009.  

Under physiological conditions, Aβ production reportedly associated with neuronal activity (Ciritto et al, NEURON, 2005 etc.) and regulated by the sleep/wake cycle (Kang et al., SCIENCE, 2009 etc.).

  1. 4.1. Antimicrobial activity of Aβ peptide and α-synuclein, please refer to the studies regrading Herpes virus, Aβ, and AD, such as Kumar, et al., SciTranslMed, 2016; Eimer et al., NEURON, 2018; Readhead et al., NEURON, 2018 etc., which is also reviewed in Itzhaki et al., FasebJ, 2017.

Response. We added a brief review of these articles in section 4.1 on page 5 and corresponding references [52-55] in the list of publication.

  1. On page 5, regarding α-syn and antimicrobial activity, please specify the aggregation status of α-syn, because many amyloid fibrils, not monomers, are capable of disturbing membranes.

Response:  Sometimes the aggregation status of proteins is not assayed in detail. We tried to add some data about this in several places. For example, in section 4.1, on page 6 we added the following text: “Although the characteristics of α-synuclein used in these assays are not described, most probably, recombinant protein was not aggregated.

“The results of this study demonstrate that native α-synuclein expression in neurons inhibits viral growth and injury in the CNS.”, “injury in the CNS” is not sure from the references.

Response: we deleted “injury in the CNS”.

  1. On page 6, “either producing soluble amyloid precursor protein-α by α-secretase or generating shorter Aβ species such as Aβ1-15 and Aβ1-16 by the subsequent cleavage with β-secretase and α-secretase.”, the α-cut of APP and the subsequent γ-cut produces p3 peptides.

Response: Section 5, containing this text was deleted, as recommended by Reviewer 1.

  1. Figure 2, the scheme of “cross-seeding” may be misleading. Currently, various cross-seeding models have been proposed and it is not sure viral amyloids cross-seed oligomers of amyloidogenic proteins/peptides. Please cite appropriate references for the cross-seeding model. In the figure, the term “exogenous” may be confusing.

Responses: a) We added the following sentence in Section 5 on page 8 :after “…CoV-2 infections and Parkinson’s disease [82]: ”Currently, various models have been proposed and further studies are needed to clarify an exact mechanism of cross-seeding”

b) The references [81, 82] is right below Figure 2 and its legend.

c) We also replaced “exogenous” on “host”.

Thanks again for reviewer’s comments and suggestions. They helped us improve the manuscript.

Round 2

Reviewer 1 Report

Authors should arrange references at the end of the list of cited works (79-86) and in the text of the article.

Author Response

We arranged references 79-86 and in the text of the article. Thank you very much

Reviewer 2 Report

The revised version has been markedly improved. There is a very small issue. In the section 6 " Relationship between COVID-19 viral proteins and amyloidogenic PPs.", the references are sometimes not properly formatted, which may be corrected before publication.

Author Response

We formatted references in the section 6. Thank you very much.